# Entrapment of *N*-Hydroxyphthalimide Carbon Dots in Different Topical Gel Formulations: New Composites with Anticancer Activity

**DOI:** 10.3390/pharmaceutics11070303

**Published:** 2019-07-01

**Authors:** Corina-Lenuta Savin, Crina Tiron, Eugen Carasevici, Corneliu S. Stan, Sorin Alexandru Ibanescu, Bogdan C. Simionescu, Catalina A. Peptu

**Affiliations:** 1Department of Natural and Synthetic Polymers, Faculty of Chemical Engineering and Environmental Protection, “Gheorghe Asachi” Technical University of Iasi, 71, Prof., Dr. Docent Dimitrie Mangeron Street, 700050 Iasi, Romania; 2Regional Institute of Oncology, 2-4, General Henri Mathias Berthelot Street, 700483 Iasi, Romania; 3“Petru Poni” Institute of Macromolecular Chemistry, 41A Aleea Grigore Ghica Vodă street, 700487 Iasi, Romania

**Keywords:** composite, *N*-hydroxyphthalimide, carbon dots, polymer gels, antitumoral activity

## Abstract

In the present study, the antitumoral potential of three gel formulations loaded with carbon dots prepared from *N*-hydroxyphthalimide (CD-NHF) was examined and the influence of the gels on two types of skin melanoma cell lines and two types of breast cancer cell lines in 2D (cultured cells in normal plastic plates) and 3D (Matrigel) models was investigated. Antitumoral gels based on sodium alginate (AS), carboxymethyl cellulose (CMC), and the carbomer Ultrez 10 (CARB) loaded with CD-NHF were developed according to an adapted method reported by Hellerbach. Viscoelastic properties of CD-NHF-loaded gels were analyzed by rheological analysis. Also, for both CD-NHF and CD-NHF-loaded gels, the fluorescence properties were analyzed. Cell proliferation, apoptosis, and mitochondrial activity were analyzed according to basic methods used to evaluate modulatory activities of putative anticancer agents, which include reference cancer cell line culture assays in both classic 2D and 3D cultures. Using the rheological measurements, the mechanical properties of gel formulations were analyzed; all samples presented gel-like rheological characteristics. The presence of CD-NHF within the gels induces a slight decrease of the dynamic moduli, indicating a flexible gel structure. The fluorescence investigations showed that for the gel-loaded CD-NHF, the most intense emission peak was located at 370 nm (upon excitation at 330 nm). 3D cell cultures displayed visibly larger structure of tumor cells with less active phenotype appearance. The in vitro results for tested CD-NHF-loaded gel formulations revealed that the new composites are able to affect the number, size, and cellular organization of spheroids and impact individual tumor cell ability to proliferate and aggregate in spheroids.

## 1. Introduction

Statistically, it has been reported that approximately 8 million people die from different types of cancer each year in the world; including breast, lung, liver, skin, and brain cancers, etc. So far, the main therapeutic methods for cancer treatment remain surgery and chemotherapy; however, researchers have made enormous efforts in the last years to develop new compounds that are better tolerated by patients during cancer therapy. In order to attack cancer cells and to reduce the same effect in the healthy cells, new treatments methods have been investigated. Among them, transdermal drug delivery systems appear as a promising alternative strategy to carry antineoplastic agents due to certain several advantages such as increased drug solubility, better bioavailability, high stability, controlled drug release, prolonged half-life, selective organ or tissue distribution, and reduction of the total required dose [1]. Enhanced topical delivery of the active principle to the target site, including those for cancer therapy, can be achieved by noninvasive drug delivery systems which can ensure sustained therapy with a single application, thus avoiding first-pass hepatic metabolism, gastric degradation, or frequent dosing and also the inconvenience of parenterals [2].

Carbon dots (CD) are a new star of the carbon nanomaterials family; their unique properties mean they are attractive materials for a wide range of applications such as bioimaging, biosensing, drug delivery, optoelectronics, photovoltaics, and photocatalysis [3,4]. Therefore, more and more researchers are paying significant interest to the synthesis, properties, and applications of these carbon nanostructured materials. However, the investigations toward drug delivery for biomedical applications are still at their beginning [5], along with their putative antitumoral properties. Precursors used to synthesize carbon dots determine their properties, creating a new opportunity for finding new anticancer molecules for certain types of cancer (personalized cancer therapy) [6]. Gels based on natural and/or synthetic polymers represent a good pathway for biomedical applications, due to their hydrophilic properties and their biocompatibility. Physical gels are usually prepared by mixing a polymer and a solvent, with or without thermal treatment, resulting a homogeneous material with remarkable mechanical properties related to the aggregation process of the polymer chains [7]. So far, different types of natural or synthetic polymers have been used to prepare polymer–drug conjugates, produced mainly by physical entrapment of the biologically active principle into macromolecular hydrogels, micro/nanoparticles, or liposomes in therapeutic approaches for the treatment of various cancers.

Polysaccharides are renewable resources found in all living organisms, which have been widely used in different important fields such as the biomedical, pharmaceutical, and food industries and tenvironmental remediation. Their unique properties, such as biocompatibility, biodegradability, nontoxicity, hydrophilicity, stability, and structural variability, allow a high capacity for carrying biological information, making them suitable as a promising natural biomaterial [8]. In recent years, the scientific community has used polysaccharides in a broad range of biomedical applications, such as drug delivery systems for the treatment of inflammation, cell–cell recognition, immune responses, metastasis (tumor cells are spreading from primary tumor to secondary organs e.g., from breast to lungs), and in tissue engineering in scaffolds and wound dressings [9]. Sodium alginate (AS) is a natural polysaccharide, extracted from brown seaweed, used in various biomedical applications such as drug delivery, tissue engineering, and wound healing, due to the attractive properties such as biocompatibility, low toxicity, ease of manipulation, and mild gelation [10]. Another attractive polysaccharide is carboxymethylcellulose (CMC) in its sodium salt form. CMC is the major cellulose ether, also known as cellulose gum, formed from carboxymethyl ether groups. CMC’s valuable properties, such as its hydrophilic nature, capacity to form gels at high concentration, and thixotropy, make it suitable for biomedical applications such as drug delivery and tissue engineering [11]. From the synthetic polymers category, carbomers (CARB) are a high-molecular-weight crosslinked polymer of acrylic acid, which are playing an important role in many commercial products such as gels, creams, and lotions, providing viscosity, stabilization, and suspension properties [12].

In this article, we focus on the embedding of *N*-hydroxyphthalimide carbon dots (CD-NHF) in different continuous matrices formed of natural (alginate or carboxymethyl cellulose) or synthetic (CARB) polymers. Due to the recently proven antitumoral activity of NHF [13], CD-NHF were developed and the composite was proven also to induce breast cancer cell apoptosis at doses that only marginally affect normal cell counterparts. The reason for embedding CD-NHF is on one hand, to reduce the aggregation tendency of CD which can affect the treatment efficacy, and on the other hand, to be protected by a polymer environment against chemical modification, which can also affect their properties. The fluorescence analysis demonstrates the CD-NHF presence within the gel. Appropriate gel properties were investigated from a rheological point of view considering different polymer concentrations; the optimum formulation has been used as the matrix for CD-NHF and further biological investigations.

Cancer is a disease that causes cells to change and grow in an uncontrolled manner. Many—if not all—of the tumor aggressivity behaviors are caused by key molecular defects in multiple biochemical pathways controlling cell survival, proliferation, differentiation, and integration in histological tisular structures and immune interactions. Apoptosis is a cell suicide program that plays an important role in tissue homeostasis by eliminating unnecessary cells [14,15]. The deregulation of the apoptotic pathway results in variety of diseases and has been shown to be involved in cancer cell resistance to conventional anticancer therapy [16]. There are two critical pathways of apoptosis—the extrinsic pathway (death receptor-mediated pathway) and the intrinsic pathway (mitochondria-mediated pathway)—and caspases are fundamental players in both pathways [17]. Mitochondria are vital organelles for energy production and intracellular Ca^2+^ homeostasis and they are involved in a variety of cellular processes, including differentiation, proliferation, and apoptosis. It has been shown that abnormal mitochondrial dynamics have an important role in tumorigenesis. Also, the increased anaerobic glycolysis activity in neoplastic cells was the result of a dysfunction of the mitochondrial activity [18].

We consider that CD-NHF incorporation in gels could be a convenient way to manipulate its local availability (and effects) and its persistence. While the actual mechanism of action is still incompletely understood for many emerging nanoformulations, interactions with extracellular matrix components are plausible objectives in many drug designs. The incorporation of CD-NHF in both gel varieties utilized in our assay was thought to be promising for possible targeted applications in certain malignancies (perhaps including basocellular and spinocellular skin carcinomas). The effect of gels with CD-NHF in 4T1 (mouse mammary breast cancer), MDA-MB-231 (human mammary breast cancer), HDMVECn (primary dermal microvascular endothelial cells), Balb/c-5064 (mouse dermal microvascular endothelial cells), A375 (human malignant melanoma), and B16F10 (mouse malignant melanoma) cells in 2D (cultured cells in normal plastic plates) and 3D (Matrigel) models was evaluated by assessing cell viability, mitochondrial activity, and apoptosis.

## 2. Materials and Methods

### 2.1. Materials

The following materials were used for the experiments: the carbomer Ultrez 10 was provided by Antibiotice SA Iasi (Iasi, Romania); *N*-hydroxyphthalimide, medium-viscosity sodium alginate (extracted from *Macrocytis pyrifera*), and glycerin (99.9%) were purchased from Sigma-Aldrich (St. Louis, MO, USA); medium-viscosity carboxymethyl cellulose from Fluka (Buchs, Switzerland); and reagent-grade ethanol (EtOH) and Milli-Q ultrapure distilled water from Merck Chemicals (Darmstadt, Germany). The chemical reagents used in this study were of analytical-grade purity and were used without further purification. MDA-MB-231 cells were purchased from the American Type Culture Collection, Rockville (ATCC), VA, USA; 4T1 cells were from the American Type Culture Collection, Rockville, MD, USA; bovine serum was purchased from Sigma-Aldrich; HDMVECn (primary dermal microvascular endothelial cells) were from ATCC; Balb/c-5064 (mouse dermal microvascular endothelial cells) were from Cell Biologics (Chicago, IL, USA); and Matrigel Matrix from BD Biosciences (356234), Bedford, MA, USA. For biological tests, the Live and Dead Cell Assay (Abcam ab115347), CellTiter-Blue^®^ Cell Viability Assay (Promega, (Madison, WI, USA), MitoTracker™ Red (Molecular Probe, Eugene, OR, USA), Matrigel (Corning, Bedford, MA, USA ), and Caspase-3/7 Assay (Promega, Madison, WI, USA ) were used according to the manufacturer’s recommendations.

### 2.2. Methods

#### 2.2.1. Preparation of Gel Formulations Based on CARB, AS, and CMC

Antitumoral gels based on CARB, AS, and CMC, respectively, and loaded with CD-NHF were prepared according to the method reported by Hellerbach et al. [19] with slight modification. CD-NHF were prepared according to a protocol reported by Stan et al. [4]. Prior to gel embedment, the carbon dots were prepared through pyrolytic processing of an NHF precursor using an experimental setup presented in Appendix A. Briefly, in a typical synthesis procedure, 0.1 g of NHF is thermally processed in carefully controlled conditions (temperature of the main sequence of the pyrolytic process is 253 °C within 30 min), leading to a typical configuration of the carbon dots consisting of a carbonaceous core with a surface highly decorated by various functional groups. The slightly different processing parameters compared with the previously reported preparation path are due to the different experimental setup, which in this work, was reconfigured and extended in order to increase both the reproducibility of the carbon dots’ preparation path and the resultant per batch quantities. The resultant reaction mass is primarily dispersed in high-purity water kept cooled at 4–5 °C. In the next stages, the suspension is centrifuged at 10,000 rpm for 15 min and the supernatant is collected and freeze-dried, with a fine powder of carbon dots being obtained. A series of gel formulations were prepared by mixing different quantities of CARB, AS, or CMC (Table 1) in a solution mixture consisting of 10 mL distilled water and 3 mL ethanol in glass vessels at room temperature. In order to achieve a uniform gel formulation, a T 18 digital ULTRA-TURRAX disperser by IKA was used (10 min, 10,000 rpm). The gelling agent’s concentration ranged from 3.8 to 5.8% (Table 1). In order to prepare the gel formulations containing CD-NHF (CARB-F4, AS-F6, or CMC-F6, respectively), the CD-NHF (3.84 mg/mL) was added into the water–ethanol solution and dispersed by sonication for 1 min at room temperature using a sonication bath (Bandelin Sonorex, Berlin, Germany), followed by adding CARB, AS, or CMC. The obtained gel formulations and CD-NHF gel (Appendix A) were stored at 4° C in a refrigerator. The basic molecular formulas for the polymers and CD-NHF used in order to obtain the gel formulations with and without CD-NHF are: sodium alginate (AS)—(C_6_H_8_O_6_)*_n_*, carboxymethyl cellulose (CMC)—[C_6_H_7_O_2_(OH)_2_OCH_2_COONa]*_n_*, Ultrez 10 carbomer (CARB)—(C_3_H_4_O_2_)*_n_*, and *N*-hydroxyphthalimide (NHF)—C_8_H_5_NO_3_.

#### 2.2.2. Characterization of the Prepared NHF Carbon Dots

The morpho-structural characteristics of the prepared NHF carbon dots were essentially similar to those previously reported [4]. In brief, the XPS survey and high-resolution O1s, N1s, and C1s spectra revealed the presence of the defect-rich graphitic core highly decorated with various terminal functional groups, while the recorded FT-IR spectra revealed the modifications/reconfigurations occurring through thermal processing of the NHF precursor and confirmed by XPS results which were revealing the presence of various functional groups attached to the graphitic core, which are essentially responsible for both optical and antitumoral properties of this type of carbon dots. In Appendix A are presented the recorded spectra of the NHF precursor and resultant carbon dots, and several specific vibrations of various groups are detailed. Interesting details regarding the morphology of the prepared carbon dots were highlighted by the HR-TEM investigation. Figure 1 presents the recorded micrographs of two samples obtained by depositing a highly diluted chloroform-dispersed carbon dot solution on 300-mesh carbon-plated copper grids. Both images suggest a cluster organization of smaller entities, which presumably are individual carbon dots. While their aspect could suggest individual carbon dots, it is also possible to be attributed to even smaller clusters, as can be seen in Figure 1b.

#### 2.2.3. Rheological Studies

Viscoelastic properties of all gel formulations were evaluated in triplicate by using an Anton Paar Physica MCR 501 (Graz, Austria) rheometer. The modular rheometer is equipped with an electronically commutated synchronous motor (ECMotor, Graz, Austria), allowing rheological testing in controlled stress (CS) and controlled strain (CR) modes. Also, the instrument allows the individual creation of complex real-time tests containing a large number of different intervals under CS or CR control, both in rotational and oscillatory modes. The rheometer is provided with an H-PTD200 system for the temperature control. A parallel-plate geometry specially designed for gels (with serrated plates to avoid slippage) with a diameter of 50 mm was chosen as the measuring system. The freshly prepared, well-homogenized gel formulations were introduced into the measuring cell and analyzed. The rheometer has an intelligent configuration system for automatic identification and configuration—“Tool Master”—which performs automatic transfer of the parameters of the measuring system (constructive characteristics, operating constants, geometry) and temperature control to the Rheoplus program. The chip integrated into the geometry contains all data connected to that and transfers them automatically to the program.

#### 2.2.4. Fluorescence Analysis

The fluorescence analysis of CD-NHF and CD-NHF-loaded gel formulations were recorded on a Horiba Fluoromax 4P (Fluor Essence Version 35.1.20) provided with the Quanta-φ integration sphere. Also, a visual testing of photoluminescence properties was performed using a Philips UVA TL4WBLB lamp (München, Germany) with the emission maximum located in the 370–390 nm range and a 50 mW, 440 nm laser diode.

#### 2.2.5. Cell Culture

MDA-MB-231 cells were cultured in F-12K medium supplemented with 100 U/mL of penicillin and 100 μg/mL of streptomycin and 5% bovine serum; 4T1 cells were cultured in RPMI-1640 supplemented with 100 U/mL of penicillin and 100 μg/mL of streptomycin and 10% bovine serum; HDMVECn (primary dermal microvascular endothelial cells) and Balb/c-5064 (mouse dermal microvascular endothelial cells) were cultured in Vascular Cell Basal Medium supplemented with Microvascular Endothelial Cell Growth Kit-BBE ATCC^®^ PCS-110-040; A375 (human malignant melanoma) and B16F10 (mouse malignant melanoma) were cultured in Dulbecco’s Modified Eagle’s Medium supplemented with 100 U/mL of penicillin and 100 μg/mL of streptomycin and 10% bovine serum.

#### 2.2.6. Cell Proliferation and Apoptosis Activity

For cell viability estimation, we used the CellTiter-Blue^®^ Cell Viability Assay (Promega). Cells were seeded into a 96-well flat-bottomed tissue culture plate at a density of 2000 cells/well and allowed to adhere to the plate by incubating at 37 °C under 5% CO_2_ overnight. Following cell attachment, the cells were incubated with the tested CD-NHF at 5% concentrations (50 µg/mL) for 72 h. For all in vitro experiments, we used the 5% CD-NHF concentration, based on our preliminary results. Control cells were treated with phosphate-buffered saline (PBS), which was equivalent to the amount of PBS used as vehicle. After each of the 72 h treatment time periods, 50 μL of cell viability solution was added to each well and the plate was reincubated for 4 h before fluorescence recording using a multiplate microplate reader (FilterMax F5, Sunnyvale, CA, USA). Apoptosis was assessed using the Caspase-3/7 Assay (Promega) according to the manufacturer’s indications.

#### 2.2.7. Mitochondrial Activity

For mitochondrial assay, MitoTracker™ Red (Molecular Probe, Eugene, OR, USA) cells were incubated for 72 h with CD-NHF and then subjected to 1 µg/mL MitoTracker and incubated for 30 min before the fluorescence of the resultant solutions was determined at 590 nm using a multiplate microplate reader.

#### 2.2.8. 3D Matrigel Assays

The 3D Matrigel assays were conducted with 1000 cells seeded in Ibidi plates between 2 layers of Matrigel (BD Matrigel Matrix, Growth Factor Reduced (BD Biosciences)) and cultured for 14 days before microscopy analysis (TissueGnostic rig, Vienna, Austria, Europe). Twelve hours post-seeding, 3D embedded cells began to be treated with gels (CARB-F2, CMC-F3, AS-F5) alone and with gels containing 5% (50 µg/mL) CD-NHF (CARB-F4, CMC-F6, AS-F6) for 72 h. After 72 h, the treatments were removed and replaced with normal 3D Matrigel medium (medium corresponding to every cell type supplemented with 2% fetal bovine serum (FBS) and 1% Matrigel). The Live and Dead Cell Assay (Abcam) was used according to the manufacturer’s instructions. Nuclei were counterstained with NucBlue Live Ready Probes Reagent (Thermo Fisher Scientific, Eugene, OR, USA). Fluorescence pictures were acquired at 20× magnification using a Zeiss Axio Observer Z1 Fluorescence Microscope from TissueGnostics rig. Single focal plane images were acquired using Tissue FAXS 4.2 software. The TissueQuest 6.0 software was used for total area segmentation analysis and to quantify the area and sum intensity of fluorescence signal for each spheroid (Appendix A).

#### 2.2.9. Statistical Analysis

GraphPad Prism was used for statistical analysis using tests stated in the figure legends. Grouped analyses were performed by *t*-test. Significance was established when *p* < 0.05.

## 3. Results and Discussion

### 3.1. Preparation of Gel Formulations

Previous studies concerning the preparation of physical gels based on natural or synthetic polymers, namely carbomers, sodium alginate, and carboxymethyl cellulose, have pointed out that these materials are nontoxic and nonhazardous and easily synthesized. We aimed to prepare different topical CD-NHF-loaded gel formulations based on polymers with antitumoral activity. Herein, physical gel formulations based on CARB, AS, and CMC were synthesized according to a slightly modified version of the method reported by Hellerbach et al. [19]. This method offers some advantages compared with other methods reported, such as being simple, rapid, and consistent; performed in the absence of crosslinkers (hence, we eliminated the possibility of toxic chemical contamination commonly associated with methods which use covalent crosslinkers); and the product is a homogeneous soft material formed at room temperature.

Different concentrations of CARB, AS, or CMC were tailored and their effects were observed. First, simple gels were prepared and analyzed before loading with CD-NHF. The gel formulations with or without CD-NHF obtained were stable, odorless, transparent, and highly viscous. Of them, the best formulations presenting a suitable consistency and the best viscoelastic properties, respectively, were considered for further study and loading with CD-NHF, namely samples CARB-F2, AS-F5, and CMC-F3. The carbon-dot-loaded gel samples were denoted as CARB-F4, AS-F6, and CMC-F6.

### 3.2. Rheology Studies

The viscoelastic nature of polymer gels plays an important role in their adhesion properties; a very important characteristic considering different biomedical applications. Gels based on natural or synthetic polymers exhibit an excellent adhesion property due to both the elastic and viscoelastic properties.

Using the rheological measurements, the mechanical properties of the best formulations were analyzed. All analyzed samples presented rheological characteristics of a gel. The obtained results showed that viscosity was directly dependent on the polymeric content of the formulations. During the rheological measurements, both dynamic moduli were constant. As can be observed from Figure 2, the storage modulus (G’) exhibited higher values than the loss modulus (G”)) over the entire strain range of 0.01 to 100%, indicating the gel-like behavior (Figure 2). The results obtained by rheological analysis confirmed that CARB-F2, AS-F5, and CMC-F3 samples were optimized and further considered for loading with CD-NHF, due to the gel equilibrium modulus. The presence of CD-NHF (for samples CARB-F4, AS-F6, and CMC-F6) in the gel structure has a significant influence over both dynamic moduli. In the case of the AS-F6 sample, the CD-NHF addition induced a slight increase of the dynamic moduli, indicating stiffness of the gel structure, whereas in CARB-F4 and CMC-F6 gels, the effect of CD-NHF addition was exactly the opposite, leading to a slight decrease of the dynamic moduli, indicating a flexible gel structure. Also, the amplitude sweep allows determination of the limits of the linear viscoelastic range for the prepared gels. These values are not affected by the presence of CD-NHF (Table 2).

All tested gels showed a solid-like behavior, as the storage modulus (G’) was always larger than the loss modulus (G”). As one can observe from Figure 2, the CARB and CMC gel formulations have a more flexible structure than the AS gel, confirmed by storage modulus (G’) values (G’_CARB_ (620 Pa) > G’_CMC_ (540 Pa) > G’_AS_ (437 Pa)). Frequency sweep tests allow observation of the elastic response of gels. The presence of the hydrogen bonding is evidenced by the frequency dependence of the moduli over the entire range of 0.05–500 1/s (γ_LVE_(CARB) > γ_LVE_(CMC) > γ_LVE_(AS)) (Figure 3).

### 3.3. Fluorescence Analysis

In order to confirm the presence of CD-NHF in the gel matrices, the formulations were further evaluated through fluorescence analysis. Fluorescence spectra exhibit the maximum emission (λ_em_) at 424 nm when using an excitation wavelength (λ_ex_) of 370 nm for CD-NHF (Figure 4 and Appendix A). Also, Figure 4 illustrates the emission profile of CD-NHF-loaded gels (CARB-F4, AS-F6, and CMC-F6) at three different excitation wavelengths ranging from 370 to 410 nm. Moreover, the recorded results (Appendix A) revealed that at 370 nm, in the case of CARB, the blue-light emission was not significantly affected by the CARB matrix. The observed difference between CD-NHF blue-light emission and CARB matrix was only 1.5% (Appendix A). When AS and CMC were used as a matrix for the entrapment of CD-NHF, an 11% decrease of the blue-light emission was observed, compared to the case of the CARB matrix and CD-NHF. Herein, we can conclude that in this case, the polymer matrix clearly plays an important role for the modification of CD-NHF optical properties.

### 3.4. In Vitro Studies

Analysis of the cytostatic or cytolytic or growing pattern modulatory activities may be done with end point assays or live cell imaging techniques. 3D cultures create a better similarity between the cultured cells and the living organism. The basic methods to evaluate modulatory activities of putative anticancer agents include reference cancer cell line cultures assays in both classic 2D and 3D cultures.

Viability of HDMVECn cells (primary dermal microvascular endothelial cells) in 2D culture treated with simple gel formulations (CARB-F2, AS-F5, and CMC-F3) and gel formulations containing CD-NHF (conc. 5%; CARB-F4, AS-F6, and CMC-F6) was not affected by CARB-F4 and CMC-F6, whereas, interestingly, AS-F6 had a significant effect (Figure 5).

Proliferation activity of melanoma cell lines (2D system) was not affected the by the presence of CARB-F2, CMC-F3, or AS-F5 formulations (Figure 6a), respectively, while cell viability was affected in a gradual manner in cells treated with CARB-F4, CMC-F6, and AS-F6 formulations (Figure 6b,c).

In the 3D Matrigel assay, the human melanoma cells form visible and consistently larger colonies compared with the same cells treated with CD-NHF-loaded gel formulations (conc. 5%) (Figure 7 and Appendix A).

By comparing the left column (Matrigel) with the median (simple gel formulations), it can be observed that spheroids are less compact, with decreased cellular uniformity and a tendency to dissociate. In the median column, we find that CMC and AS gel formulations are more unfavorable for maintaining a homogeneity of metabolic activity (the green shade in the L/D kit reflecting live cell population) and induce (or amplify) the dissociation tendency of cells from differentiated proliferation aggregates. The differences found in the analysis of 3D spheroids can be caused by the fact that the different types of gel formulations used either provide different stability conditions for extracellular molecules that favor maintaining a differentiated cellular phenotype, or employ different receptors for adhesion to cell membranes favorable to maintaining both differentiation and inhibition of the cellular dissemination tendency [20,21]. Comparing the median column (single gel formulations) with that on the right (gel formulations supplemented with 5% CD-NHF), we find that exposure to CD was unfavorable to cell proliferation, with global cell counts being significantly impaired and the quantitative weight of cells in invasion reduced. We specify that the shade of the green marker is proportional (L/D kit) to the level of cell metabolic activity. Smaller color intensities can be determined by both a decreased number of cells and more restricted metabolic activity in CD-NHF exposure. Also, the morphological aspect of spheroids under the influence of gel formulations containing CD-NHF is presented in Figure 8.

Also tested was the viability potential of two different breast cancer cell lines, namely 4T1 (mouse breast cancer) and MDA-MB-231 (human breast adenocarcinoma), in 2D and 3D culture models. The viability of the breast cancer cell lines 4T1 (mouse breast cancer) and MDA-MB-231 (human breast adenocarcinoma) in the 2D culture system treated with CARB-F4 was affected (Figure 9a). Mitochondrial activity was affected upon CARB-F4 treatment (Figure 9b). In the 3D Matrigel assay, the malignant cells form visible and consistently larger colonies compared with same cells treated with CARB-F4 (Figure 9c). Moreover, the apoptotic foci in 3D cultures treated with gels containing CD-NHF were significantly higher (Figure 9c; white squares in boxes 2 and 4). The apoptotic pattern in the whole spheroid body is also distorted: while in native culturing, inner cells (typically associated with hypoxic conditions inside large spheroids) in spheroids began the apoptotic program, at 5% CD-NHF in exposed spheroids, this form of cell death is initiated in all spheroid compartments. 3D cell cultures displayed visibly larger structure of cancer cells with reduced active phenotype appearance. This suggests that treatment with the CARB-F4 formulation could also either induce cell senescence or cell dormancy—a topic that deserves further clarification. Together, these data suggest that CD-NHF-loaded gels affect the number, size, and cellular organization of spheroids and impacts individual cancer cell ability to proliferate and aggregate in spheroids.

## 4. Conclusions

This paper reported the development via a physical method of three topical gel formulations loaded with CD-NHF, which were evaluated from the point of view of the modulatory activities of putative anticancer agents. The obtained results indicate that CD-NHF-loaded gels matrices present complex and interesting cell modulatory activities which are relevant for cancer control in potential clinical applications. The mechanical properties of gels were investigated, concluding that the presence of CD-NHF in the gels induced a slight decrease of the dynamic moduli, indicating a more flexible gel structure. The fluorescence analysis confirmed the presence of the embedded CD-NHF in all gel formulations. Also, worth noting is the potential antitumoral activity of the gel formulations loaded with CD-NHF. The most important finding is that the CD-NHF-loaded gel formulations are able to affect the number, size, and cellular organization of spheroids, while also having a significant impact on the individual tumor cell’s ability to proliferate and aggregate in spheroids.

## Figures and Tables

**Figure 1 pharmaceutics-11-00303-f001:**
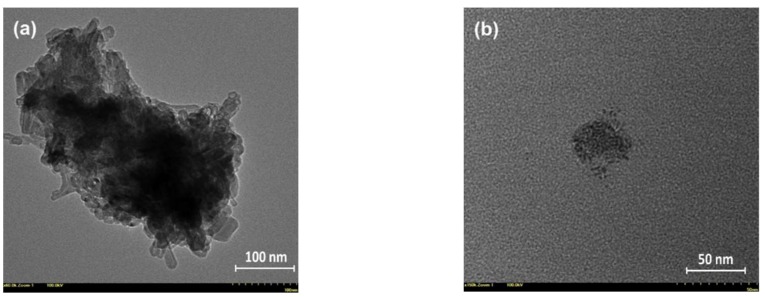
Recorded HR-TEM images of the carbon dots prepared from *N*-hydroxyphthalimide (CD-NHF): (**a**) CD-NHF with an average size of the clusters in 100–300 nm range and (**b**) CD-NHF with an average size of the smaller entities in 2–4 nm range

**Figure 2 pharmaceutics-11-00303-f002:**
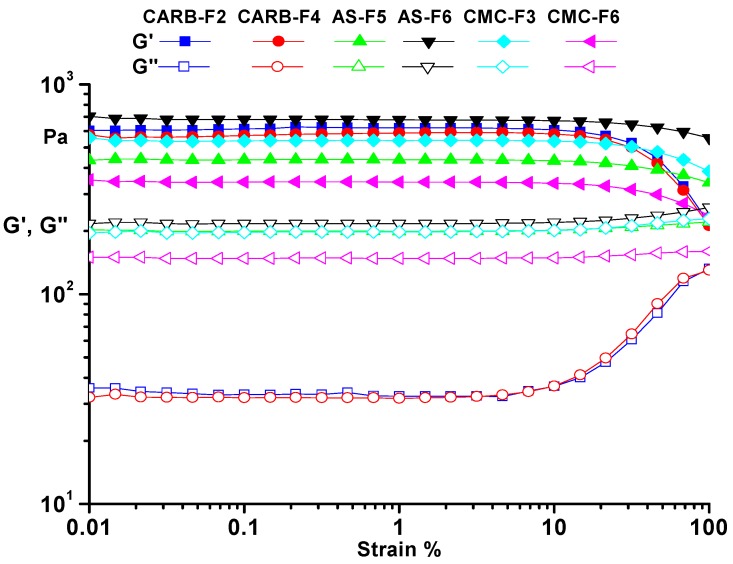
Amplitude sweep for simple gels (CARB-F2, AS-F5, and CMC-F3) and gels with CD-NHF (CARB-F4, AS-F6, and CMC-F6). G’: storage modulus; G”: loss modulus.

**Figure 3 pharmaceutics-11-00303-f003:**
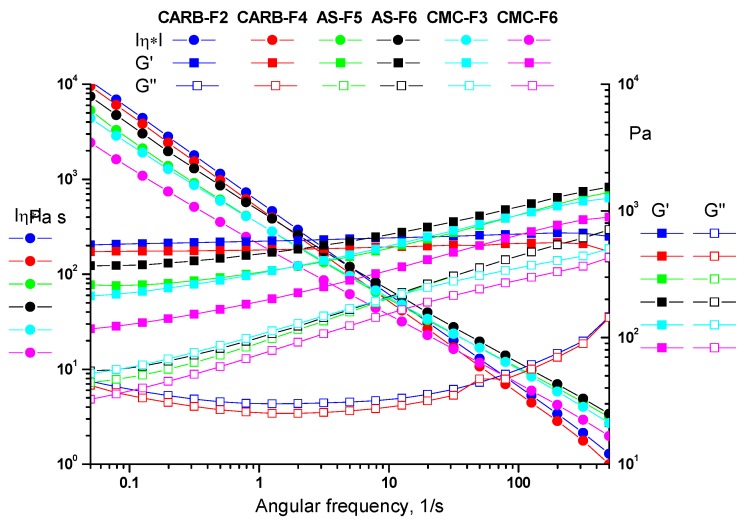
Frequency sweep for simple gels (CARB-F2, AS-F5, and CMC-F3) and gels with CD-NHF (CARB-F4, AS-F6, and CMC-F6).

**Figure 4 pharmaceutics-11-00303-f004:**
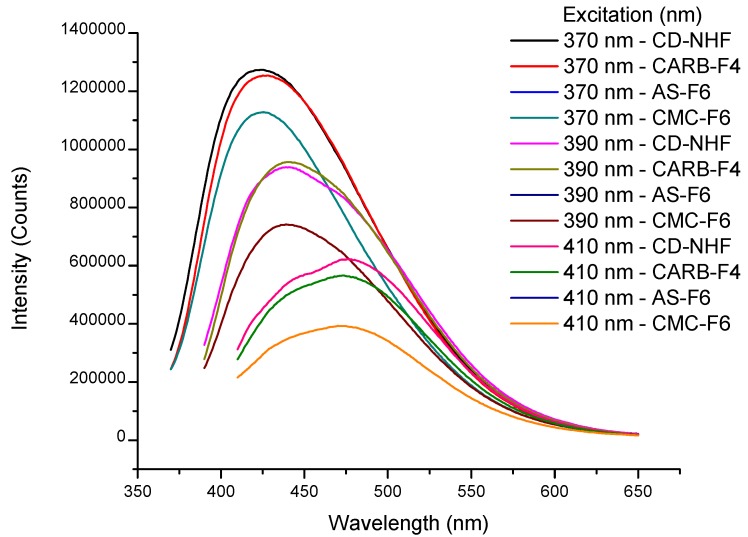
Emission spectra of CD-NHF suspended in H_2_O: CARB-F4, AS-F6, and CMC-F6 samples.

**Figure 5 pharmaceutics-11-00303-f005:**
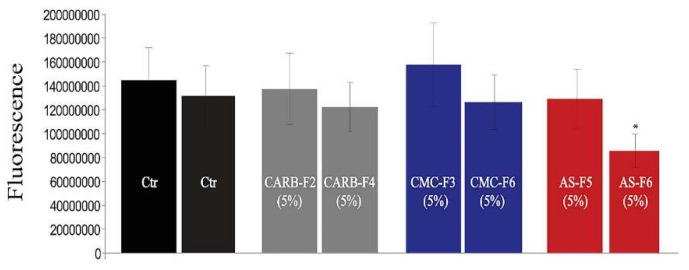
Viability of normal dermal cells (Primary Dermal Endothelial Cells (PDMC)) for gel formulations without and with CD-NHF (*N* = 15 wells/column from two independent experiments). Ctr (Control).

**Figure 6 pharmaceutics-11-00303-f006:**
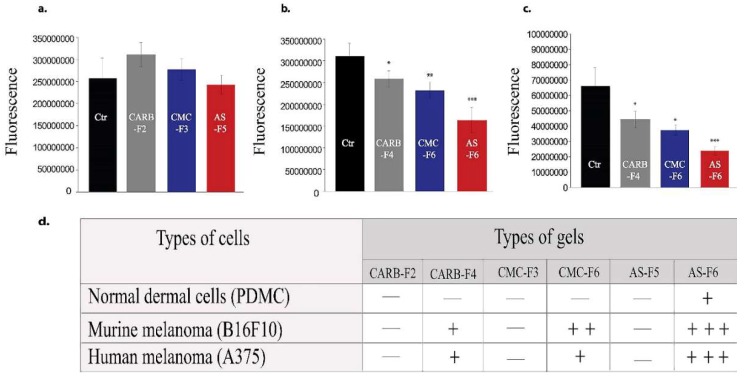
Murine and human melanoma cell viability for gel formulations without (**a**) and with (**b**,**c**) CD-NHF (conc. 5%). *N* = 15 wells/column from two independent experiments; viability of treated groups is expressed relative to the control group. (**a**) Skin cancer; (**b**) murine melanoma (B16F10); (**c**) human melanoma (A375); (**d**) scoring of significance effect, where +++ (***) *p* = 0.0003, ++ (**) *p* = 0.01, + (*) *p* = 0.04, and — = no effect.

**Figure 7 pharmaceutics-11-00303-f007:**
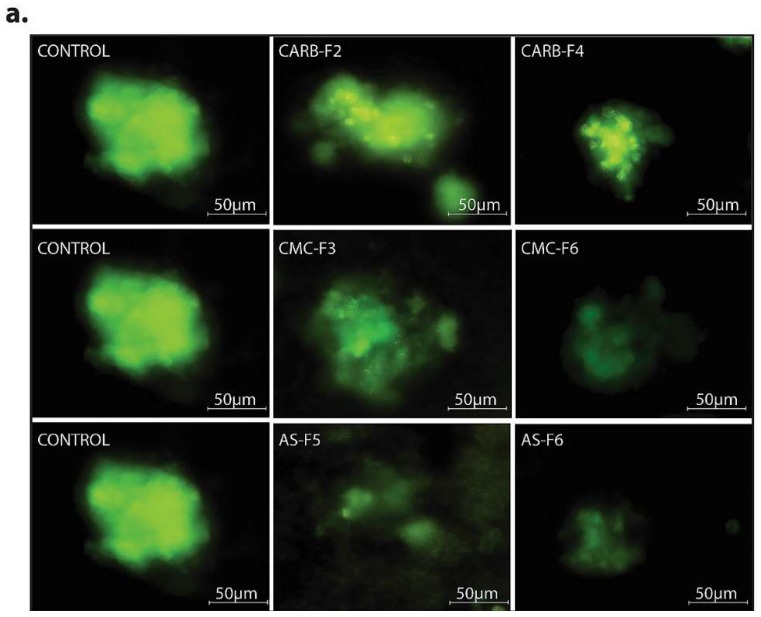
(**a**) 3D Matrigel assay for human melanoma cell cultures in standard conditions (left) and in the presence of added CD-NHF-free gel (middle) versus 5% CD-NHF-loaded gel (right). *N* = 3D Matrigel cultures per type of gel with or without CD-NHF; 20× microscope objective. (**b**) Sum intensity of fluorescence signal; (**c**) area of spheroids expressed in μm^2^.

**Figure 8 pharmaceutics-11-00303-f008:**
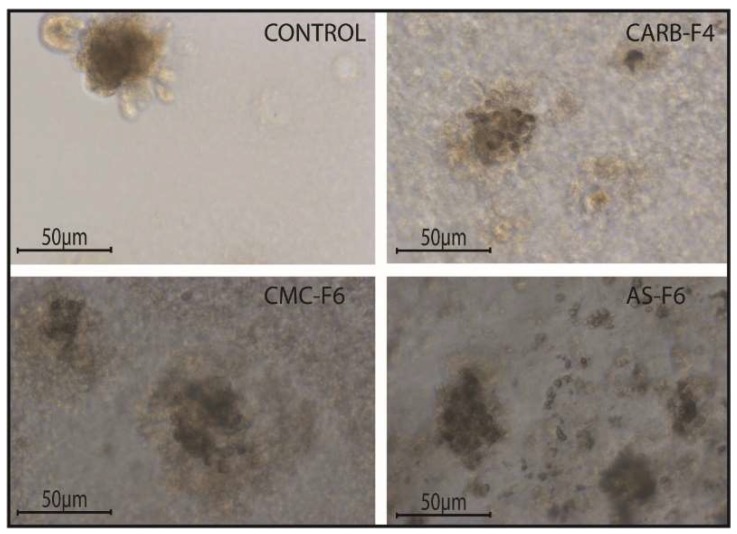
The morphological aspect of spheroids under the influence of gel formulations. *N* = 3D Matrigel cultures per type of gel with or without CD-NHF; 20× magnification.

**Figure 9 pharmaceutics-11-00303-f009:**
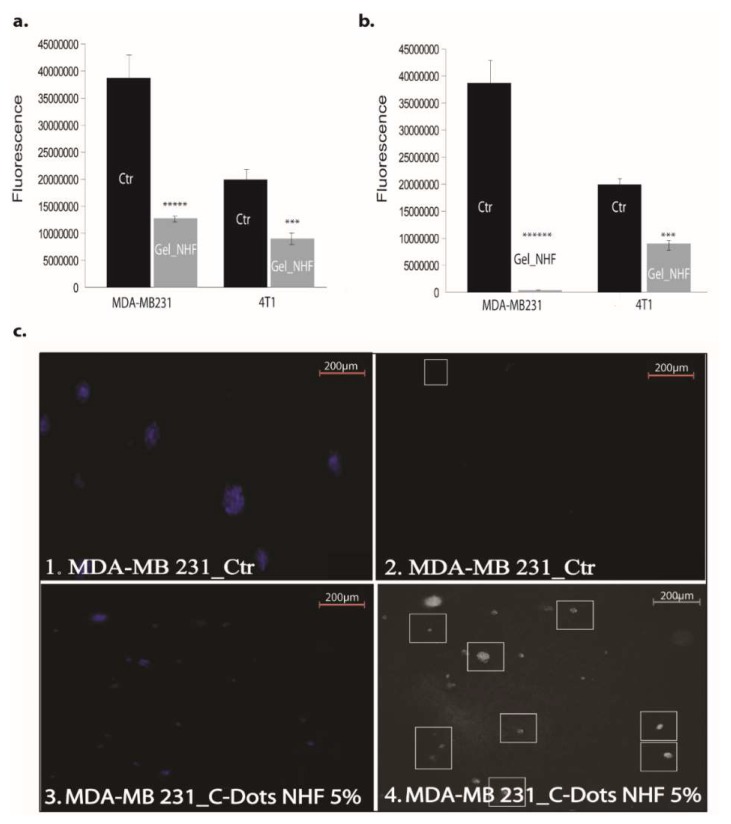
The effects of gels with 5% CD-NHF in 2D and 3D culture. (**a**) Viability of MDA-MB-231 (human breast adenocarcinoma 33.3%) and 4T1 (mouse mammary breast cancer 44.87%) cell lines treated with gels containing 5% CD-NHF (*N* = 8 wells/column from two independent experiments); (**b**) mitochondrial activity in MDA-MB-231 (human breast adenocarcinoma 2%) and 4T1 (mouse mammary breast cancer 49.2%) treated with gels containing 5% CD-NHF (*N* = 8 wells/column from two independent experiments); (**c**) apoptosis evidence in MDA-MB-231 by immunofluorescence microscopy and 3D Matrigel culture treated with gels containing 5% CD-NHF (*N* = 6; 3 Matrigel culture controls and 3 Matrigel cultures treated with 5% NHF). (c1,c3) represent NucBlue nuclei staining; (c2,c4) reflect apoptosis. Picture magnification 5×; where *** *p* < 0.0005, ***** *p* < 0.000005, ****** *p* < 0.0000005.

**Table 1 pharmaceutics-11-00303-t001:** Polymeric composition of various CARB, AS, and CMC gel formulations. CARB: carbomer Ultrez 10; AS: sodium alginate; CMC: carboxymethyl cellulose; GLY: glycerin.

Sample Code	CARB, %	AS, %	CMC, %	GLY, %	Distilled Water (mL)	Ethanol (mL)	CD-NHF (g)
CARB-F1	3.8	-	-	-	10	3	-
CARB-F2	4.6	-	-	-			-
CARB-F3	5.8	-	-	-			-
CARB-F4	4.6	-	-	-			0.050
AS-F1	4.6	4.6	-	-			-
AS-F2	4.6	4.6	-	9.2			-
AS-F3	3.5	4.6	-	8.1			-
AS-F4	2.3	4.6	-	6.9			-
AS-F5	1.2	4.6	-	5.8			-
AS-F6	4.6	4.6	-	5.8			0.050
CMC-F1	2.9	-	2.9	-			-
CMC-F2	2.9	-	2.9	5.8			-
CMC-F3	3.6	-	2.9	6.5			-
CMC-F4	4.3	-	2.9	7.2			-
CMC-F5	5.8	-	2.9	8.7			-
CMC-F6	3.6	-	2.9	6.5			0.050

**Table 2 pharmaceutics-11-00303-t002:** The limits of the linear viscoelastic range (γ_LVE_) for the tested gel formulations.

Sample Code	γ_LVE_ (%)
CARB-F2	0.25
CARB-F4	0.25
AS-F5	5
AS-F6	5
CMC-F3	10
CMC-F6	10

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
