# Peer review of "Entrapment of N-Hydroxyphthalimide Carbon Dots in Different Topical Gel Formulations: New Composites with Anticancer Activity"

_pharmaceutics, 2019, doi:10.3390/pharmaceutics11070303_

Round 1
Reviewer 1 Report
This manuscript illustrates the entrapment of carbon dots into three different types of gel. Also, the carbon dots-loaded gel exhibited antitumor activities. The work was carefully made. However, I don't understand if you have tested the antitumor activities of carbon dots alone. If carbon dots have already exhibited the antitumor activities, why do you embed carbon dots into three gel to reduce the efficacy? In addition, there are several mistakes to be addressed.
English grammar and style mistakes. For example, in the title, you'd better keep all the letters in the uniform format, namely either capitalize all the first letter of each word or keep all of them lower case except the first word. In abstract line 25, please rewrite the sentence "rheological and fluorescence properties points of view". In line 35, "impacts" should be "impact". In line 45, the word "cell" is repeated. Please delete unnecessary "cell".
Please check the requirement of the journal. Usually there are five key words.
In line 59-60, "Hence, researchers have made enormous efforts to develop new efficient strategies based on natural products for cancer therapy" has no relation with the sentence before.
The content from line 77 to 88 is suggested to be combined. It applies also to the content from line 100-112.
In line 89, carbon dots "are" not "is".
In line 93, "However, the investigations toward drug delivery for biomedical application are still at the beginning" needs references to support. For example, "Carbon dots: promising biomaterials for bone-specific imaging and drug delivery, Nanoscale, 9 (2017) 17533-17543.", "Crossing the blood-brain barrier with nanoparticles, Journal of Controlled Release, 207 (2017) 290-303".
In line 141, the unit of temperature needs to be checked.
Table 1 shows many sample codes. What is the difference between them?
In line 166, photoluminescent should be changed to photoluminescence.
What technique did you use to obtain Figure 7? Confocal? If so, do you think it possible to measure and compare the light intensities of all the three columns?
Line 283-286 needs references.
Generally in the reference section, I didn't find sufficient references.
Author Response
Dear Editor-in-Chief, dear reviewer,
We are very grateful to the reviewer for his suggestions. We considered all of them and the modifications can be now seen in the new revised version of the manuscript (all modifications were colored in yellow).
Response to Reviewer 1 Comments
Comments and Suggestions for Authors
1. This manuscript illustrates the entrapment of carbon dots into three different types of gel. Also, the carbon dots-loaded gel exhibited antitumor activities. The work was carefully made. However, I don't understand if you have tested the antitumor activities of carbon dots alone. If carbon dots have already exhibited the antitumor activities, why do you embed carbon dots into three gel to reduce the efficacy? In addition, there are several mistakes to be addressed.
Answer: Carbon dots alone have been tested on several concentrations (1%, 3%, 5% and 10%) on several normal and cancer cell lines (human and murine) like HMLE, MDA-MB-231, 4T1, L363, A375, BEAS-2B, A549, RPMI8226, HT29 and U87, showing their anticancer activity, but the results are the subject of a different work. (Tiron CE et al, Imide derived Carbon Dots – A New Promising Approach in Cancer Treatment, Journal of Drug Delivery Science and Technology, submitted 2019). Based on these results we tested in the present manuscript only 5% CD-NHF entrapped in gel because this CD-NHF concentration do not affect normal cells but affect in different manner cancer cells of several cell line type; e.g. do not affect glioma U87 cells.
The reason for embedding CD-NHF is on one hand to reduce the aggregation tendency of CD which can affect the treatment efficacy and on the other hand to be protected by polymer environment against chemical modification which can affect their properties. Also, CD-NHF incorporation in gels could be convenient way to manipulate its local availability (and effects) and its persistence. While the actual mechanism of action is still incomplete understood for many emerging nano formulates, interactions with extracellular matrix components per se are plausible objectives in many drug designs. Both gel incorporation and precisely those 2 varieties utilised in our assay were thought to be promising for possible targeted applications in certain malignancies (perhaps including basocellular and spinocellular skin carcinomas).
2. English grammar and style mistakes.
Answer: The entire manuscript has been reviewed
For example, in the title, you'd better keep all the letters in the uniform format, namely either capitalize all the first letter of each word or keep all of them lower case except the first word.
Answer: The title was modified in a uniform format all first letters were modified with lowercase.
3. In abstract line 25, please rewrite the sentence "rheological and fluorescence properties points of view".
Answer: The sentence has been rewritten: ”Viscoelastic properties of CD-NHF-loaded gels were analyzed by rheological analysis. Also, for both CD-NHF and CD-NHF-loaded gels the fluorescence properties were analyzed.
4. In line 35, "impacts" should be "impact".
Answer: The word was modified.
5. In line 45, the word "cell" is repeated. Please delete unnecessary "cell".
Answer: The repetitive word was deleted.
6. Please check the requirement of the journal. Usually there are five key words.
Answer: The five keywords are: composite, N-hydroxyphthalimide, carbon dots, polymer gels, cancer, antitumoral activity
7. In line 59-60, "Hence, researchers have made enormous efforts to develop new efficient strategies based on natural products for cancer therapy" has no relation with the sentence before.
Answer: The entire Introduction section has been changed and re-organized.
8. The content from line 77 to 88 is suggested to be combined. It applies also to the content from line 100-112.
Answer: The content was modified at the reviewer suggestion.
9. In line 89, carbon dots "are" not "is".
Answer: The word was modified at the reviewer suggestion.
10. In line 93, "However, the investigations toward drug delivery for biomedical application are still at the beginning" needs references to support. For example, "Carbon dots: promising biomaterials for bone-specific imaging and drug delivery, Nanoscale, 9 (2017) 17533-17543.", "Crossing the blood-brain barrier with nanoparticles, Journal of Controlled Release, 207 (2017) 290-303".
Answer: the reviewer is right, the reference has been added in the text
11. In line 141, the unit of temperature needs to be checked.
Answer: The unit of temperature has been changed.
12. Table 1 shows many sample codes. What is the difference between them?
Answer: In this study we consider three types of polymeric matrices (CARB: carbomer Ultrez 10; AS: sodium alginate; CMC: carboxymethyl cellulose) in which polymer concentration was varied in order to obtain appropriate gel properties from rheological point of view. This is the main difference between different formulations showed in the mentioned table.
13. In line 166, photoluminescent should be changed to photoluminescence.
Answer: The modification has been done.
14. What technique did you use to obtain Figure 7? Confocal? If so, do you think it possible to measure and compare the light intensities of all the three columns?
Answer: The technique was explained in detail on the 2.2.7. The 3D Matrigel assays presented in the Methods part. Light intensity has been measured and represented in main text and figure 7, which include now also the quantification data expressed as charts.
15. Line 283-286 needs references.
Answer: References were added. It’s about the fact that components of gels alone can affect stiffness of extracellular matrix and cell behavior.
16. Generally, in the reference section, I didn't find sufficient references.
Answer: Extra references were added.
Sincerely yours,
Catalina A. Peptu (PhD)
Reviewer 2 Report
The authors systematically investigates the loading of CD-NHF into different gels and applied these different gel formulations for anti-cancer studies in both 2D and 3D cultures. It is a rather interesting work. However, some important information are missing and the figures need to be re-organized in a more professional manner for easy comparison.
1. Since CD-NHF is the active ingredient in the gel formulation, its synthesis procedure should be briefly described. The authors should also provide some basic material characterization results such as TEM or absorbance, fluorescence spectra. What is the biological effect of the CD-NHF if not entrapped in gel?
2. Page 4, line 141: Please amend the typo of 40oC. It should be 4oC.
3. Please resize and re-organize the Figure 1-4.
4. The authors should take the photos of as-formed gels with or without CD-NHF under white light and UV light. It is also necessary to provide the basic molecular formula of the gel monomers.
5. The data presented in Figure 5 & 6 should be further processed to show cell viability %.
6. Figure 4a&b can be combined in 1 graph to compare the cell viability with or without CD-NHF to clearly show the effect of the dots.
7. Please check the labeling of Figure 9. The cell types are labeled wrongly.
8. I think it is necessary to construct a table comparing the properties and the anti-tumor performances of all the gels made. What are their advantages and limitations?
Author Response
Dear Editor-in-Chief, dear reviewer,
We are very grateful to the reviewer for his suggestions. We considered all of them and the modifications can be now seen in the new revised version of the manuscript (all modifications were colored in yellow).
Response to Reviewer 2 Comments
Comments and Suggestions for Authors
The authors systematically investigate the loading of CD-NHF into different gels and applied these different gel formulations for anti-cancer studies in both 2D and 3D cultures. It is a rather interesting work. However, some important information is missing and the figures need to be re-organized in a more professional manner for easy comparison.
Since CD-NHF is the active ingredient in the gel formulation, its synthesis procedure should be briefly described. The authors should also provide some basic material characterization results such as TEM or absorbance, fluorescence spectra. What is the biological effect of the CD-NHF if not entrapped in gel?
--Answer: the reviewer is right and consequently the description in brief of CD-NHF has been introduced in the manuscript (the photo of laboratory installation for CD-NHF synthesis has been added in Supplementary materials) together with TEM analysis, FTIR. The fluorescence spectrum was in the manuscript but now the figures are better organized.
Carbon dots alone have been tested on several concentrations (1%, 3%, 5% and 10%) on several normal and cancer cell lines (human and murine) like HMLE, MDA-MB-231, 4T1, L363, A375, BEAS-2B, A549, RPMI8226, HT29 and U87, showing their anticancer activity, but the results are the subject of a different work. (Tiron CE et al, Imide derived Carbon Dots – A New Promising Approach in Cancer Treatment, Journal of Drug Delivery Science and Technology, submitted 2019). Based on these results we tested in the present manuscript only 5% CD-NHF entrapped in gel because this CD-NHF concentration do not affect normal cells but affect in different manner cancer cells of several cell line type; e.g. do not affect glioma U87 cells.
Page 4, line 141: Please amend the typo of 40oC. It should be 4oC.
--Answer: The temperature unit has been modified.
Please resize and re-organize the Figure 1-4.
--Answer: The figures 1-4 were re-organized.
The authors should take the photos of as-formed gels with or without CD-NHF under white light and UV light. It is also necessary to provide the basic molecular formula of the gel monomers.
--Answer: The basic molecular formula for used polymer in order to obtain gels were included in the manuscript at section 2.2.1. Preparation of gel formulations based on CARB, AS and CMC presented in the Methods part. The photos of as-formed gels with or without CD-NHF under white light and UV light were added in Supplementary materials figure S3.
The data presented in Figure 5 & 6 should be further processed to show cell viability %.
--Answer: Fixed, the percent viability was added in figure legend only were resulted in significant effect, to keep the figure simple.
Figure 4a&b can be combined in 1 graph to compare the cell viability with or without CD-NHF to clearly show the effect of the dots.
--Answer: Fixed, we think referee means figure 5 as figure 4 do not contain cell viability.
Please check the labeling of Figure 9. The cell types are labeled wrongly.
--Answer: Added viability percentage and supplementary explanations about figure 9c 1,2, 3 and 4.
I think it is necessary to construct a table comparing the properties and the anti-tumor performances of all the gels made. What are their advantages and limitations?
--Answer: A new table was made and included in the manuscript. Gels CMC-F3 and AS-F5 alone affected normal cell viability and from this reason we extended investigations only with gel CARB-F2. Fig 6 updated with requested table.
Sincerely yours,
Catalina A. Peptu (PhD)
Reviewer 3 Report
Recommendation: Not suitable for publishing in Pharmaceutics.
In this manuscript authors reported the preparation of composites of polymeric gels loaded with N-hydroxyphthalimide carbon dots. I express serious reservations on several aspects of the manuscript. In overall, authors are not clear with their concepts, design of studies, and presentation of the data.
Major Concerns:
1. Introduction- Authors presented irrelevant information in the introduction that is nowhere connected to the main theme of the study. Title of the manuscript is “Entrapment of N-hydroxyphthalimide Carbon dots in different topical gel formulations–new composites with antitumoral activity”. The explanation for the topical formulations, the antitumor component is missing the entire manuscript except in the title. Why authors choose carbon dots for the study and the advantages of making such composite materials is missing. In other words, what are the issues with carbon dots or polymer gel formulations and how one can overcome these issues with composites is missing? Moreover, antitumor is generally used for the in vivo tumor models and anticancer is opt for cell line-based studies. The introduction is full of irrelevant, random and inadequate citations.
2. Methods- Methods in the manuscript must include quantities and all the steps involved in the preparation and characterization of materials to ensure reproducibility. Though the N-hydroxyphthalimide Carbon dots were prepared using reported procedure, methodology and spectroscopic and particle characterization of materials must be included in the manuscript and should be compared with its carbon dot loaded polymeric gel formulations. Correct the mistakes like “40 ˚C in the refrigerator” “Balb/c-5064 instead of Babb/c-5064” in proof-reading.
3. Results and discussion: Although synthesis of Carbon dots is not a novelty of the manuscript, proper characterization of carbon dots is missing in the manuscript. For example, FT-IR spectra, DLS and TEM or AFM images of Carbon dots must be included. Moreover, the complete characterization of these composite gel materials with and without doping with carbon dots must be included.
It would be interesting to see the dose-dependent cell viability of these gels in the in-vitro cell culture studies. Moreover, the release of carbon dots from the gels, uptake, and enrichment in the cells must be done.
Results from the graphs were not explained properly, in other words, use quantitative terms such as “fold increase or decrease or percentage changes” rather than the words used in the manuscript (for example less, more, much better, much diminished etc.). It would be more informative if the graphs under In vitro studies were plotted in terms of percentage cell viability rather than fluorescence.
Moreover, fluorescent images of the 3D-cultured cells were of not quality and there is no counterstain to see the cell population. Quantitative measurements will tell more than talking in terms of fluorescence.
Minor concerns:
1. Sentences and paragraphs were poorly formatted. In several sections, sentences were incomplete and fused with the next lines, were not making any sense.
2. Irrelevant and random citations were reported throughout the manuscript. An adequate number of recent papers on preparation and biological activates of carbon dots were not cited.
Author Response
Dear Editor-in-Chief, dear reviewer,
We are very grateful to the reviewer for his suggestions. We considered all of them and the modifications can be now seen in the new revised version of the manuscript (all modifications were colored in yellow).
Response to Reviewer 3 Comments
Comments and Suggestions for Authors
In this manuscript authors reported the preparation of composites of polymeric gels loaded with N-hydroxyphthalimide carbon dots. I express serious reservations on several aspects of the manuscript. In overall, authors are not clear with their concepts, design of studies, and presentation of the data.
Major Concerns:
1. Introduction- Authors presented irrelevant information in the introduction that is nowhere connected to the main theme of the study. Title of the manuscript is “Entrapment of N-hydroxyphthalimide Carbon dots in different topical gel formulations–new composites with antitumoral activity”. The explanation for the topical formulations, the antitumor component is missing the entire manuscript except in the title. Why authors choose carbon dots for the study and the advantages of making such composite materials is missing. In other words, what are the issues with carbon dots or polymer gel formulations and how one can overcome these issues with composites is missing?
Answer: The introduction section has been entirely re-written and re-organized in conformity with reviewer queries. New paragraphs has been introduced adding explanations regarding the ”topical formulations, the antitumor component”. Also in the title the word ”antitumoral” has been replaced with ”anticancer” considering the reviewers suggestion.
2. Why authors choose carbon dots for the study and the advantages of making such composite materials is missing. In other words, what are the issues with carbon dots or polymer gel formulations and how one can overcome these issues with composites is missing?
Answer: Due to the recently proved antitumoral activity of NHF (xx)_, CD-NHF were developed and the composite was proven also to induce the breast cancer cells apoptosis at doses that only marginally affect normal cell counterparts. The reason for embedding CD-NHF is on one hand to reduce the aggregation tendency of CD which can affect the treatment efficacy and on the other hand to be protected by polymer environment against chemical modification which can also affect their properties. The fluorescence analysis demonstrates the CD-NHF presence within the gel. C-Dots incorporation in gels could be convenient way to manipulate its local availability (and effects) and its persistence. While the actual mechanism of action is still incomplete understood for many emerging nanoformulates, interactions with extracellular matrix components per se are plausible objectives in many drug designs. Both gel incorporation and precisely those 2 varieties utilised in our assay were thought to be promising for possible targeted applications in certain malignancies (perhaps including basocellular and spinocellular skin carcinomas).
3. Moreover, antitumor is generally used for the in vivo tumor models and anticancer is opt for cell line-based studies.
Answer: Also in the title the word ”antitumoral” has been replaced with ”anticancer” considering the reviewers suggestion.
4. The introduction is full of irrelevant, random and inadequate citations.
Answer: The introduction section has been entirely re-written and re-organized in conformity with reviewer queries.
5. Methods- Methods in the manuscript must include quantities and all the steps involved in the preparation and characterization of materials to ensure reproducibility. Though the N-hydroxyphthalimide Carbon dots were prepared using reported procedure, methodology and spectroscopic and particle characterization of materials must be included in the manuscript and should be compared with its carbon dot loaded polymeric gel formulations.
Answer: the reviewer is right and consequently the description in brief of CD-NHF has been introduced in the manuscript (the photo of laboratory installation for CD-NHF synthesis has been added in Supplementary materials) together with TEM analysis, FTIR. The fluorescence spectrum was in the manuscript but now the figures are better organized. Also, the photos of as-formed gels with or without CD-NHF under white light and UV light were added in Supplementary materials figure S3.
6. Correct the mistakes like “40 ˚C in the refrigerator” “Balb/c-5064 instead of Babb/c-5064” in proof-reading.
Answer: All mistakes concerning the temperature and Balb/c-5064 were corrected.
7. Results and discussion: Although synthesis of Carbon dots is not a novelty of the manuscript, proper characterization of carbon dots is missing in the manuscript. For example, FT-IR spectra….TEM or AFM images of Carbon dots must be included. Moreover, the complete characterization of these composite gel materials with and without doping with carbon dots must be included.
Answer: answer related to answer no 5.
8. It would be interesting to see the dose-dependent cell viability of these gels in the in-vitro cell culture studies. Moreover, the release of carbon dots from the gels, uptake, and enrichment in the cells must be done.
Answer: We do not have dose-dependent cell viability studies of these gels. As we have mentioned, we have previously tested carbon dots alone for several concentrations (1%, 3%, 5% and 10%), and on several normal and cancer cell lines (human and murine) like HMLE, MDA-MB-231, 4T1, L363, A375, BEAS-2B, A549, RPMI8226, HT29 and U87. (Tiron CE et al, submitted 2019). We do not have data regarding the release of carbon dots from the gels, uptake, and enrichment in the cells but carbon dots entrapped in gel exhibited similar antitumoral effect as carbon dots alone.
9. Results from the graphs were not explained properly, in other words, use quantitative terms such as “fold increase or decrease or percentage changes” rather than the words used in the manuscript (for example less, more, much better, much diminished etc.).
Answer: percentage was used for fluorescence analysis.
10. It would be more informative if the graphs under In vitro studies were plotted in terms of percentage cell viability rather than fluorescence.
Answer: Fixed, the percent viability was added in figure legend only were resulted in significant effect, to keep the figure simple.
11. Moreover, fluorescent images of the 3D-cultured cells were of not quality and there is no counterstain to see the cell population. Quantitative measurements will tell more than talking in terms of fluorescence.
Answer: Fixed by adding in fig S4 the acquired microscopic pictures displaying separate staining channels of color (green for cell viability, blue as nuclear counterstaining), and merged images. Supplementary figure S5 displays the analytical segmentation procedure for a typical spheroid image. Fluorescent pictures were acquired using a conventional, standard Zeiss Axio Observer Z1 Microscope in single focal plane. Under these circumstances, the software quantification of the viability figure for a target spheroid cumulates fluorescence from cells situated on the focus plane (where microscope capture maximum of fluorescent intensities) and from surrounding planes (resulting in more blurred signals). While unavailable here, a confocal microscope would provide better accurate images from inside of 3D spheroids.
Fig 7 was modified to include charts of fluorescence measurements for intensities and area.
Minor concerns:
1. Sentences and paragraphs were poorly formatted. In several sections, sentences were incomplete and fused with the next lines, were not making any sense.
Answer: Sentences and paragraphs were formatted.
2. Irrelevant and random citations were reported throughout the manuscript. An adequate number of recent papers on preparation and biological activates of carbon dots were not cited.
Answer: Citations were formatted.
Sincerely yours,
Catalina A. Peptu (PhD)
Round 2
Reviewer 2 Report
The authors have carefully answered my comments. Only a minor suggestion: the figure captions of Figure 5 to 7 are too long. The authors should state what are no. 1-7 as legends in the figure.
Author Response
Dear Editor-in-Chief, dear reviewer,
We would like to thank to the Reviewer for reviewing our manuscript so thoughtful. We have carefully taken the comments into consideration in preparing our new revision. All the modifications from the new revised version of the main manuscript, are now clearly highlighted, using the "Track Changes". The English style of the paper has been revised carefully. In the following we answer to the reviewer comments, point by point.
Response to Reviewer 2 Comments
The authors have carefully answered my comments. Only a minor suggestion: the figure captions of Figure 5 to 7 are too long. The authors should state what are no. 1-7 as legends in the figure.
Answer
We agree with the reviewer suggestion and consequently figures 5, 6 and 7 were modified accordingly to the main manuscript, pages 11, 12 and 14.
Sincerely yours,
Catalina A. Peptu (PhD)

Reviewer 3 Report
I appreciate the authors for addressing the majority of the questions in the revised manuscript. I still have some reservations to improve the presentation quality of the manuscript. It can be published after following corrections without the need for further review.
Introduction becomes lengthy for article format; some portions can be cut off from the introduction such as cancer, its pathways, mitochondria, apoptosis, and their mechanisms, aerobic glycolysis. These portions are nowhere connected to the present theme of the manuscript, scientific and experimental approaches.
Authors should use quantitative terms (such as percentages, fold changes) for the explanation of results and in the discussion sections rather qualitative terms that they used (such as less, more, much better, much diminished).
I still find some spelling mistakes, few meaningless complex sentences. Authors should revise with extensive and thorough editing of the English in the manuscript before the formal acceptance.
Author Response
Dear Editor-in-Chief, dear reviewer,
We would like to thank to the Reviewer for reviewing our manuscript so thoughtful. We have carefully taken the comments into consideration in preparing our new revision of the manuscript. All the modifications from the new revised version of the main manuscript, are now clearly highlighted, using the "Track Changes". In the following we answer to the reviewer comments, point by point.
Response to Reviewer 3 Comments
I appreciate the authors for addressing the majority of the questions in the revised manuscript. I still have some reservations to improve the presentation quality of the manuscript. It can be published after following corrections without the need for further review.
Introduction becomes lengthy for article format; some portions can be cut off from the introduction such as cancer, its pathways, mitochondria, apoptosis, and their mechanisms, aerobic glycolysis. These portions are nowhere connected to the present theme of the manuscript, scientific and experimental approaches.
Answer
We agree with the reviewer assessment concerning the introduction length which was partially modified. However, we want to mention that the paragraph mentioned from the introduction which is explaining the terms like “cancer, its pathways, mitochondria, apoptosis, and their mechanisms, aerobic glycolysis” has correspondence in our experimental approach namely figure 9b, 9c and paragraph from line 355-370 which highlights mitochondrial and apoptosis investigations. Therefore, we cannot totally exclude that paragraph from the introduction.
Authors should use quantitative terms (such as percentages, fold changes) for the explanation of results and in the discussion sections rather qualitative terms that they used (such as less, more, much better, much diminished).
Answer
The explanation of results from the discussion section was reorganized and modified.
I still find some spelling mistakes, few meaningless complex sentences. Authors should revise with extensive and thorough editing of the English in the manuscript before the formal acceptance.
Answer
The English style of the paper has been revised carefully.
Sincerely yours,
Catalina A. Peptu (PhD)
